# IL-6 Responsiveness of CD4^+^ and CD8^+^ T Cells after Allogeneic Stem Cell Transplantation Differs between Patients and Is Associated with Previous Acute Graft versus Host Disease and Pretransplant Antithymocyte Globulin Therapy

**DOI:** 10.3390/jcm11092530

**Published:** 2022-04-30

**Authors:** Tor Henrik Anderson Tvedt, Stefan Rose-John, Galina Tsykunova, Aymen Bushra Ahmed, Tobias Gedde-Dahl, Elisabeth Ersvær, Øystein Bruserud

**Affiliations:** 1Department of Hematology, University of Oslo, 0424 Oslo, Norway; tgeddeda@ous-hf.no; 2Section for Hematology, Institute of Clinical Science, University of Bergen, 5007 Bergen, Norway; oystein.bruserud@helse-bergen.no; 3Section for Hematology, Department of Medicine, Haukeland University Hospital, 5021 Bergen, Norway; galina.tsykunova@helse-bergen.no (G.T.); abah@helse-bergen.no (A.B.A.); 4Institute of Biochemistry, Kiel University, Olshausenstrasse 40, 24118 Kiel, Germany; rosejohn@biochem.uni-kiel.de; 5Institute of Clinical Medicine, University of Oslo, 0315 Oslo, Norway; 6Department of Biomedical Laboratory Scientist Education, Western Norway University of Applied Sciences, 5063 Bergen, Norway; elisabethersver@hvl.no

**Keywords:** interleukin 6, STAT3, mTOR, T cells, allogeneic stem cell transplantation, graft-versus-host disease

## Abstract

Graft-versus-host disease (GVHD), one of the most common and serious complications after allogeneic stem cell transplantation, is mediated by allocative T cells. IL-6 mediates both pro- and anti-inflammatory effects and modulates T cell response through classical signaling and trans-signaling. We investigated the effects on the mTOR and JAK/STAT pathways after various types of IL-6 signaling for circulating T cells were derived from 31 allotransplant recipients 90 days post-transplant. Cells were stimulated with IL-6 alone, hyper-IL-6 (trans-signaling), IL-6+IL-6 receptor (IL-6R; classical + trans-signaling) and IL-6+IL-6R+soluble gp130-Fc (classical signaling), and flow cytometry was used to investigate the effects on phosphorylation of AKT (Thr308), mTOR (Ser2442), STAT3 (Ser727) and STAT3 (Tyr705). CD3^+^CD4^+^ and CD3^+^C8^+^ T cells responded to classical and trans IL-6 stimulation with increased STAT3 (Tyr705) phosphorylation; these responses were generally stronger for CD3^+^CD4^+^ cells. STAT3 (Tyr705) responses were stronger for patients with previous acute GVHD; CD3^+^CD4^+^ cells from GVHD patients showed an additional STAT3 (Ser727) response, whereas patients without acute GVHD showed additional mTOR (Ser2448) responses. Furthermore, treatment with antithymocyte globulin as a part of GVHD prophylaxis was associated with generally weaker STAT3 (Tyr705) responses and altered STAT3 (Ser727) responsiveness of CD3^+^CD4^+^ cells together with increased mTOR (Ser2448) responses for the CD3^+^CD8^+^ cells. Thus, early post-transplant CD3^+^CD4^+^ and CD3^+^ CD8^+^ T cell subsets differ in their IL-6 responsiveness; this responsiveness is modulated by antithymocyte globulin and differs between patients with and without previous acute GVHD. These observations suggest that allotransplant recipients will be heterogeneous with regard to the effects of post-transplant IL-6 targeting.

## 1. Introduction

Inhibition of IL-6-initiated signaling through the downstream JAK/STAT pathway has emerged as a possible strategy to prevent and treat graft-versus-host disease (GVHD) [1]. Two studies suggest that combination of IL-6 blockade with standard GVHD prophylaxis reduces the rate of severe acute GVHD without increasing the rates of complications [2,3]. Janus kinase (JAK1/2) inhibition is now regarded as a therapeutic strategy for severe acute and chronic GVHD; it seems to be effective both as acute GVHD prophylaxis, in the treatment of steroid-refractory GVHD, and in salvage therapy for chronic GVHD [4,5,6,7,8,9,10]. Furthermore, animal studies suggest that a primary effect of IL-6 in acute GVHD is enhanced differentiation of pro-inflammatory T cell subsets (Th1 and Th17 cells) together with reduced development of regulatory T cells (Tregs) [11,12,13]. IL-6 activates various signaling pathways, including the JAK/STAT, PI3K/AKT/MTOR (also a main pathway downstream to antigen-recognizing T cell receptors (TCRs)) and MAPK/ERK pathways [14], but JAK/STAT signaling seems responsible for most IL-6 effects in GVHD since STAT3 activation induces the development of alloreactive Th17 cells and suppresses the development of inhibitory Tregs [15,16,17]. Taken together, these studies suggest that IL-6 inhibition is at least partly responsible for the therapeutic effect of the JAK1/2 inhibitor ruxolitinib in GVHD.

The anti-inflammatory IL-6 effects seem to be mediated mainly by the membrane-bound IL-6 receptor (IL-6R; termed classical IL-6 signaling), whereas pro-inflammatory effects are mainly mediated by the soluble IL-6/IL-6R complex (termed trans-signaling) [18]. Both types of signaling depend on the ubiquitously expressed glycoprotein 130 (gp130) for transmembrane transduction [18]. The balance between pro- and anti-inflammatory effects is mainly determined by the level of IL-6, the degree of IL-6R shedding (i.e., level of soluble IL-6R, sIL-6R) and the level of a soluble dimeric version of gp130 (sgp130), which functions as a decoy receptor for the IL-6/sIL-6R complex and thereby blocks IL-6 trans-signaling. The downstream STAT3 effects are especially difficult to predict because they depend on STAT3 isoforms (i.e., the α isoform and the shorter β isoform), the balance between various dimers/tetramers, post-transcriptional modifications (e.g., phosphorylation of STAT3 (Tyr705), which is present in both α and β isoforms, and STAT3 (Ser727), which occurs only in the α isoform) and intracellular STAT3 compartmentalization [19].

IL-6 has both pro- and anti-inflammatory effects [1], but pro-inflammatory effects seem to dominate early after allotransplantation as indicated by the correlation between the capacity of IL-6 to induce STAT3 phosphorylation in CD4^+^ T cells on day +22 post-transplant and the later development of acute GVHD [20]. It should also be emphasized that IL-6 is not only important for immunoregulation; several organs depend on IL-6 for tissue renewal and regeneration, especially the liver and gut, which are target organs in acute GVHD [21,22].

The effects of classical IL-6 signaling and trans-signaling differ between T cell subsets [1]. First, cell surface IL-6R expression depends on T cell development: naïve and effector memory T cells express IL-6R, whereas most other T cell subsets only express gp130 [23,24]. Second, IL-6R shedding differs between T cell subsets: naïve CD4^+^ T cells lose sIL-6R expression during acute inflammation, whereas CD8^+^ T cells do not express the proteases needed for this shedding [24]. Third, naïve T cells lose IL-6R expression during activation and differentiation and thereby become dependent on IL-6 trans-signaling [23]. However, possibly the most important attribute of IL-6 is related to the observation that human T cell differentiation can be initiated not only by soluble IL-6 or IL-6/gp130, but also by dendritic cells with IL-6 bound to cell surface IL-6R [25]. Thus, IL-6 is an important regulator of T cell reactivity.

Taken together, the previous studies reviewed above have demonstrated IL-6 receptor ligation initiates downstream JAK/STAT signaling, and both IL-6 inhibition but especially JAK1/2 inhibition can be used in the treatment of GVHD. In this context, we investigated downstream signaling (i.e., phosphorylation of STAT3, AKT, mTOR) in response to IL-6R ligation on circulating T cells derived from allogeneic stem cell transplant recipients [14,26] and whether differences in IL-6 responsiveness were associated with acute GVHD or previous treatment with antithymocyte globulin (ATG).

## 2. Materials and Methods

### 2.1. Patients and Collection of Samples

The study was approved by the local Ethics Committee (REK VEST 2013/634; REK VEST 2015/1410; REK Vest 2017/350). Patients seen at Oslo University Hospital (The National Hospital) and Haukeland University Hospital from 1 September 2016 until 1 August 2018 were included after written informed consent. All samples were collected at scheduled routine consultations at day +90 post-transplant. Venous blood was collected into ACD tubes (BD Vacutainer^®^, Franklin Lakes, NJ, USA), and peripheral blood mononuclear cells (PBMCs) were isolated by density gradient separation (Lymphoprep; Nycomed, Oslo, Norway). Cells were prepared and cryopreserved within 24 h and stored in liquid nitrogen. The clinical outcomes investigated included the occurrence of GVHD requiring high-dose steroid treatment before and after day +90, treatment-related mortality and relapse. Acute GVHD was diagnosed according to generally accepted criteria [27] after evaluation based on the Glucksberg score, and patients requiring intravenous treatment with ≥1 mg/kg/day of methylprednisolone (or equivalent steroid dose) were considered to have severe (i.e., grade 2–4) acute GVHD.

### 2.2. Flow-Cytometric Analysis of STAT3, AKT and mTOR Protein Phosphorylation

Cells were thawed rapidly, washed, resuspended in RPMI 1640 (ThermoFisher, Waltham, MA, USA) and allowed to rest at 37 °C in a humidified atmosphere of 5% CO_2_ for one hour. Thereafter, cells were divided into two tubes, washed twice with ice cold phosphate-buffered saline (PBS) and allowed to rest for an additional 20 min on ice. Cells in one tube were incubated with T cell activating anti-human CD3 (UCHT1, BD Biosciences, Franklin Lakes, NJ, USA) and anti-CD28 (28.1 BD Biosciences) for 15 min and then incubated with polyclonal goat anti-mouse antibodies (BD Biosciences) for 15 min. These antibodies were not added to the other tube.

Cells from each of the two initial tubes were then divided into 6 different tubes each (350–450 × 10^3^ cells/tube); cells in each tube were washed and resuspended in 100 μL ice-cold PBS. One of the six tubes was left untreated, and one of the following solutions was added to each of the five remaining tubes (final concentrations stated): (i) Hyper IL-6 5 ng/mL (provided by Stephan Rose-John), (ii) IL-6 20 ng/mL (PeproTech, Rocky Hill, NJ, USA), (iii) IL-6 20 ng/mL plus 40 ng/mL IL-6R (PeproTech Rocky Hill, NJ, USA), (iv) IL-6 20 ng/mL, 40 ng/mL IL-6R and 500 ng/mL sgp130FC (BioTech Minneapolis, MN, USA), and (v) 12-O-tetradecanoylphorbol-13-acetate (PMA) 100 μg/mL (Sigma-Aldrich, St. Louis, MO, USA). While Hyper IL-6, IL-6, sIL-6R and sgp130FC were dissolved in X-Vivo 10^®^ (Lonza, Basel, Switzerland), PMA was dissolved in DMSO. The final volume was 150 µL, and the total amount of DMSO added was 1.5 µL. The cells were then incubated in a water bath at 37 °C for 10 min before formaldehyde was added (final concentration 1.6%). After the addition of formaldehyde, the cells were washed twice, permeabilized using ice-cold methanol and kept at −80 °C for at least 24 h at this temperature. The cells were then washed twice with staining buffer and incubated with the following antibodies for 1 h: BV421 conjugated anti-Stat3-phospho-S727 (49/pstat3; only present in the α isomer), PE-Cy7 conjugated anti-mTOR-phospho-S2448 (MRRBY), Alexa647 conjugated anti-STAT3-phospho-705 (49/p-Stat3; expressed both by α and β isomers) and PE conjugated anti-AKT-phospho-473 (M89-61). Live/dead exclusion was performed using Alexa Fluor 700-conjugated cleaved-(ASP214)-PARP (F21-852). Cell surface markers were FITC-conjugated anti-CD3 (SK7), PerCP-Cy5.5 conjugated anti-CD8 (SK1) and BV510 conjugated anti-CD4 (L200). The PE-Cy7-conjugated anti-mTOR-phospho-2448 antibody was supplied by ThermoFisher; all the other antibodies were purchased from BD Biosciences.

The samples were analyzed using the BD FACSVerse flow cytometer from BD Bioscience, and the gating strategy is presented in Figure 1.

### 2.3. Flow-Cytometric Analysis of IL-6R Expression and Intracellular Cytokine Levels

Cryopreserved cells were thawed and incubated overnight in X-Vivo 10 medium^®^. Cells were thereafter washed and incubated (30 min, 4 °C) with PE-conjugated anti-IL-6R (M5), PerCP-Cy5.5-conjugated anti-CD8 (SK1), FITC-conjugated anti-CD3 (SK7) and BV510-conjugated anti-CD4 (L200) (all from BD Biosciences).

For analysis of cytokine expression, cells were incubated for 4 h in medium with or without 2 μL/mL of Leukocytes Activation Cocktail plus BD GolgiPlug containing PMA/Ionomycin/Brefeldin A (BD Biosciences); 4 μg/mL of DNase was also added (Sigma-Aldrich, Saint-Louis, MS). The cells were then washed in ice-cold PBS and stained with Near-IR Live/Dead Stain (ThermoFisher) before being washed twice with ice-cold 1% bovine serum albumin (BSA)/PBS, thereafter incubated with 200 μg/mL of Fc-receptor blocking immunoglobulin (Octagam, Octapharma Ltd., Coventry, UK) and finally fixed with 4% formaldehyde. The cells were thereafter permeabilized using Perm/Wash Buffer (BD Biosciences), incubated (60 min, 4 °C) with mouse monoclonal Alexa Fluor 647-conjugated anti-human IL17-A (N49-653, BD Biosciences), PerCP-conjugated mouse anti-human CD3 (SK7), PE-conjugated anti-CD8 (RPA-T8), PeCy7-conjugated anti-CD4 (RPA-T4) and BV510-conjugated anti-Interferon-γ (M-A251) (all from BD Biosciences). The samples were finally washed twice in permeabilization buffer before analysis using the BD FACSVerse flow cytometer from BD Bioscience; at least 20,000 lymphocytes were counted for each sample.

### 2.4. Flow Cytometry and Statistical Analyses

Statistical analyses were performed using GraphPad Prism 5 (GraphPad Software, Inc., San Diego, CA, USA). Spearman’s correlation for bivariate samples was used for correlation analyses, continuous variables were compared using non-parametric tests (Kruskal–Wallis one-way analysis of variance/Mann–Whitney-U test/Wilcoxon signed-rank test). Chi-Square tests and Fisher’s exact tests were used to compare categorized variables. Overall survival was calculated using the Kaplan–Meier product limit method. Differences were regarded as statistically significant when *p*-values were <0.05.

## 3. Results

### 3.1. Patient and Donor Characteristics

Peripheral blood samples derived from 31 allotransplant recipients were investigated (Table 1). All patients were in complete hematological remission at the time of transplantation. Fifteen patients developed acute GVHD requiring treatment with high-dose steroids (i.e., grade II gastrointestinal, or grade III/IV GVHD) before the blood sampling on day +90 post-transplant.

### 3.2. Stimulation with Anti-CD3^+^Anti-CD28 Alone Increases AKT/mTOR Phosphorylation of Circulating CD3^+^CD4^+^ and CD3^+^CD8^+^ Post-transplant T Cells

We investigated the effects of anti-CD3^+^anti-CD28 alone (referred to as TCR activation) on intracellular mediator phosphorylation of circulating T cells for the 31 transplant recipients. The results from a typical experiment are shown in Figure 1. TCR activation caused a significant increase in the phosphorylation of AKT (Thr308) and mTOR (Ser2448) for CD3^+^CD4^+^ T cells. A significant increase in AKT (Thr308) phosphorylation was also observed for circulating CD3^+^CD8^+^ T cells, whereas the mTOR (Ser2448) phosphoresponse reached only borderline significance for this cell subset. The phosphorylation of STAT3 (Ser727) that is present only on the STAT3 α isomer, and the STAT3 (Tyr705) that is present both on the α and β isomers [13], was not significantly altered for any of the two T cell subset. Finally, TCR activation did not have any effect on CD3^−^ peripheral blood mononuclear cells (data not shown).

### 3.3. PMA Increases Phosphorylation of STAT3 (Ser727), AKT (Thr308) and mTOR (Ser2448) for CD4^+^ and CD8^+^ T cells but Increases STAT3 (Tyr705) Phosphorylation Only for CD8^+^ T Cells

We investigated the effect of stimulation with PMA alone and PMA in combination with TCR activation (i.e., antiCD3^+^anti-CD28) on the phosphorylation of AKT (Thr308), mTOR (Ser2448), STAT3 (Ser727) and STAT3 (Tyr705) for the CD3^+^CD4^+^, CD3^+^CD8^+^ and CD3^−^ PBMC subsets. PMA alone caused a significant increase in the phosphorylation of AKT (Thr308), mTOR (Ser2448) and STAT3 (Ser727) for both T cell subsets and even CD3^−^ cells (Table 2, Figure 2).

PMA combined with TCR activation (anti-CD3^+^anti-CD28) also caused a highly significant increase in the phosphorylation of AKT (Thr308), mTOR (Ser2448) and STAT3 (Ser727) for both T cell subsets, but an additional increase in STAT3 (Tyr705) phosphorylation of borderline significance was observed only for CD3^+^CD8^+^ cells after combined PMA+TCR stimulation (Table 2). Thus, the CD4^+^ and CD8^+^ T cell subset differed only with regard to STAT3 (Tyr705) responsiveness.

### 3.4. Strong T Cell AKT (Thr308), mTOR (Ser2448) and STAT3 (Ser727) Phosphoresponses to PMA Are Seen Especially for Allotransplant Recipients with Previous Acute GVHD

We compared the effects of PMA stimulation alone on the CD3^+^CD4^+^, CD3^+^CD8^+^ and CD3^−^ PBMC subsets for patients with and without previous acute GVHD before day +90 post-transplant (Table 2, Figure 2). Patients with previous GVHD showed generally stronger phosphoresponses (i.e., higher statistical significance) for AKT (Thr308), mTOR (Ser2448) and STAT3 (Ser727) after stimulation with PMA alone; this was true for all three cell subsets. In contrast, patients without previous GVHD showed relatively strong responses for all three cell subsets only for mTOR (Ser2448), while the AKT (Thr308) and STAT3 (Ser727) responses were usually less significant for CD3^+^CD8^+^ cells. Finally, STAT3 (Tyr705) phosphorylation was not significantly altered by PMA stimulation for any patient subset.

The phosphoresponse pattern to PMA+anti-CD3^+^anti-CD28 showed similarities to the responses to PMA alone (Table 2, Figure 2). The responses of AKT (Thr308), mTOR (Ser2448) and STAT3 (Ser727) were generally stronger for patients with previous acute GVHD; i.e., usually highly significant responses for all three phosphosites and all three cell subsets. Finally, STAT3 (Tyr705) showed no significant responses to PMA combined with TCR activation.

To conclude, circulating T cells as well as CD3^−^ cells derived from allotransplant recipients with previous acute GVHD showed generally stronger PMA phosphoresponses for AKT (Thr308), mTOR (Ser2448) and STAT3 (Ser727), than T cells from patients without previous GVHD.

### 3.5. IL-6 Classical and Trans-Signaling Increase STAT (Tyr705) Phosphorylation of Resting CD3^+^CD4^+^ T Cells from Day +90 Post-Transplant

We investigated the effects of four different forms of IL-6 stimulation on CD3^+^CD4^+^ T cells: IL-6 alone (classical IL-6 signaling), hyper-IL-6 (trans-signaling), IL-6 plus sIL-6R (classical and trans-signaling) and IL-6+sIL-6R+sgp130FC (classical Il-6 signaling; trans-signaling blocked by sgp130FC). We first investigated their effects on AKT (Thr308), mTOR (Ser2448), STAT3 (Ser727) and STAT3 (Tyr705) phosphorylation in resting cells derived from all 31 patients. All four IL-6 signals caused highly significant increases in STAT3 (Tyr705) phosphorylation, a phosphosite present on both α and β STAT3 isoforms [13]. IL-6 stimulation did not alter the phosphorylation of AKT (Thr308), mTOR (Ser2448) and STAT3 (Ser727); the only exception was IL-6+sIL-6R+sgp130FC, which caused an increase in STAT (Ser727) phosphorylation but only of borderline significance. Thus, IL-6 classical and trans-signaling have opposite effects to PMA on resting T cells; IL-6 increases STAT3 (Tyr705) phosphorylation, whereas PMA increases phosphorylation of the three other phosphosites.

### 3.6. IL-6 Phosphoresponses of CD3^+^CD4^+^ T Cells Are Altered by TCR Activation: The STAT3 (Tyr705) Response Is Maintained While STAT (Ser727) and mTOR (Ser2448) Responses Increase

We analyzed the effects of IL-6 stimulation during TCR activation (Table 3, Figure 3). The overall results for the 31 patients showed significantly increased STAT3 (Tyr705) phosphorylation in response to all four IL-6 signals (i.e., classical and trans-signaling) in the presence of TCR activation; this is similar to the effects on resting T cells. However, additional effects were also observed during activation; IL-6, hyper-IL6 and IL-6+sIL-6R (i.e., both classical and trans-IL6 signaling) significantly increased STAT3 (Ser727) phosphorylation, whereas IL-6+sIL-6R and IL-6+sIL6R+sgp130FC increased mTOR (Ser2448) phosphorylation (possibly through their common classical signaling). Thus, IL-6 signaling had more extensive effects in activated than resting T cells.

### 3.7. IL-6 Phosphoresponses of TCR Activated CD3^+^CD4^+^ Cells from Patients with and without Previous Acute GVHD; Similar STAT3 (Tyr705) but Different STAT3 (Ser727) and mTOR (Ser2448) Responses

We compared the IL-6 responsiveness for patients with and without previous acute GVHD when anti-CD3^+^anti-CD28 was present during IL-6 stimulation. Both patient subsets showed similar and highly significant increases in STAT3 (Tyr705) phosphorylation in response to both classical and trans IL-6 signaling (Table 3). Furthermore, only patients with previous GVHD showed a significant increase in STAT3 (Ser727) phosphorylation in response to trans-signaling (hyper-IL-6), whereas only patients without GVHD showed highly significant (i.e., *p* < 0.01) increases in mTOR (Ser2448) phosphorylation in response to IL-6+sIL-6R (classical and trans-signaling) and IL-6+sIL-6R+sgp130FC (blockade of trans-signaling) (Table 3, Figure 4 left). Thus, post-transplant T cells derived from patients with and without previous acute GVHD show similar STAT3 (Tyr727) phosphoresponses to IL-6, whereas STAT3 (Ser727) and mTOR (Ser2448) responses differ between the two patient subsets.

### 3.8. IL-6 Responsiveness of Post-Transplant CD3^+^CD8^+^ T Cells Increases during T Cell Activation

We investigated the effects of various forms of IL-6 stimulation on intracellular signaling in resting CD3^+^CD8^+^ T cells for all 31 patients (Table 4). Only STAT3 (Tyr705) phosphorylation showed highly significant increases, and these effects seemed to depend mainly on IL-6-trans-signaling (hyper-IL-6, IL-6+sIL-6R). The other mediator/IL-6 combinations did not show significant effects, or the effects reached only borderline significance (*p* = 0.03).

We then analyzed the overall effects of IL-6 stimulation during TCR activation (i.e., anti-CD3^+^anti-CD28) (Table 4). The responsiveness of activated cells was generally stronger than for the resting cells. Both mTOR (Ser2448) and STAT3 (Tyr705) phosphorylation increased significantly in response to both classical and IL-6 trans-signaling (i.e., for all four IL-6 stimulations). STAT3 (Ser727) phosphorylation also increased significantly after classical and trans IL-6 stimulation.

### 3.9. IL-6 Responsiveness of CD3^+^CD8^+^ T Cells: Only Resting but Not TCR Activated T Cells from Patients with and without Previous Acute GVHD Differ in Their IL-6 Responsiveness

We compared the IL-6 phosphoresponses for patients with and without previous acute GVHD. IL-6 signaling had only minor effects in the absence of TCR activation; increased STAT3 (Tyr705) signaling in response to trans-signaling (i.e., hyper-IL-6) for patients with previous GVHD was the only exception, showing a highly significant difference in resting cells. Furthermore, no differences of high statistical significance were observed for activated CD3^+^CD8^+^ T cells either; increased STAT3 (Tyr727) phosphorylation in response to IL-6 trans-signaling (i.e., hyper-IL-6) and increased mTOR (Ser2448) phosphorylation, especially in response to IL-6+sIL-6R+sgp130FC, were observed for both patient subsets (Table 4). Thus, activated CD3^+^CD8^+^ T cells showed no major differences in IL-6 responsiveness when comparing patients with and without previous acute GVHD; this is in contrast to the CD3^+^CD4^+^ T cell subset that differed in STAT (Ser727) and mTOR (Ser2448) responses between the two patient subsets.

### 3.10. The IL-6 Phosphoresponsiveness of Mononuclear CD3^−^-Cells

The investigated CD3^−^ mononuclear cells constitute a heterogeneous cell population including B cells, NK cells and monocytes. The mediator phosphorylation in circulating CD3^−^ mononuclear cells was examined; the only highly significant IL-6 response was increased STAT (Tyr705) phosphorylation by hyper-IL-6 (i.e., trans-signaling). Two other responses reached only borderline significance when analyzing the overall results, and patients with and without previous acute GVHD showed only a few minor differences of borderline significance (Appendix A). Thus, although IL-6 exposure has relatively weak effects on circulating CD3^−^ mononuclear cells derived from allotransplant recipients, there is a difference between patients with and without previous acute GVHD also for these cells.

### 3.11. Effects of Prophylactic ATG on IL-6 Responsiveness of Circulating Post-Transplant T Cells

There was no significant association between previous acute GVHD and previous ATG prophylaxis, and we therefore compared the IL6 responsiveness of post-transplant cell subsets for patients with and without GVHD prophylaxis including ATG. The IL-6 responsiveness of CD3^+^CD4^+^ T cells showed no major differences between patients with and without previous ATG treatment, but minor differences were seen for STAT3 (Ser727) phosphorylation, which was increased in resting T cells (hyper-IL-6/trans-signaling) for ATG-treated patients but was increased in TCR-activated T cells for patients without ATG therapy (trans-signaling) (Appendix A).

ATG reduced the CD3^+^CD8^+^ T cell population that showed IL-6 induced increase in mTOR (Ser2448) phosphorylation during TCR activation (see Table 4); this mTOR response did not differ between patients with and without previous acute GVHD (Appendix A). The CD3^+^CD8^+^ cells also showed minor differences for STAT3 (Ser727) phosphorylation that was increased in resting T cells after ATG (classical and trans-signaling) and increased in activated T cells for patients not receiving ATG (trans-signaling).

The T cell responses to PMA also differed between patients with and without ATG therapy. IL-6 responsiveness was generally weaker for patients with previous ATG therapy with regard to AKT (Thr308), mTOR (Ser2448) and STAT3 (Ser727) phosphorylation, and the strong STAT3 (Tyr705) phosphoresponse to PMA alone for CD3^+^CD4^+^ cells was not detected after ATG (Appendix A).

Finally, we compared the phosphoresponsiveness of CD3^−^ mononuclear cells; the only highly significant difference was the increased STAT3 (Tyr705) phosphorylation in response to hyper-IL-6 (i.e., trans-signaling) for patients with previous ATG therapy (Appendix A).

Taken together, these observations suggest that ATG prophylaxis alters the IL-6 responsiveness of T cells, especially CD3^+^CD8^+^ T cells, and thereby alters the post-transplant immunoregulation.

### 3.12. Circulating T Cells from Patients with and without Previous Acute GVHD Differ in the Levels of Circulating Th1 but Not in Th2 Levels, Th17 Levels and T Cell Expression of IL-6R

Levels of circulating CD3^+^CD4^+^ and CD3^+^CD8^+^ T cells did not differ between patients with and without previous acute GVHD (Table 5): the same was true for Th2 and Th17 cells, whereas a significantly higher proportion of circulating Th1 cells was observed in patients without previous acute GVHD. Finally, we did not detect any difference in IL-6R expression when comparing circulating T cells and non-T cells derived from patients with and without previous acute GVHD (Table 5).

### 3.13. The IL-6 Responsiveness of day +90 Post-Transplant T Cells Does Not Predict the Probability to Later Wean off Systemic Immunosuppression

Follow-up information was available only for the first 12 months post-transplant; 14 patients had then developed chronic GVHD that required systemic immunosuppressive therapy (Table 1). The IL-6 responsiveness to classical and IL-6-trans-signaling of CD3^+^CD4^+^ and CD3^+^CD8^+^ T cells on day +90 post-transplant showed no association with later development of chronic GVHD requiring immunosuppressive therapy one year post-transplant (data not shown).

## 4. Discussion

We investigated how circulating CD4^+^ and CD8^+^ T cells derived from allotransplant recipients 90 days post-transplant responded to cis and trans IL-6 stimulation, and we investigated both resting and activated T cells (i.e., PMA or TCR stimulation). Our present results suggest that early post-transplant CD3^+^CD4^+^ and CD3^+^ CD8^+^ T cell subsets differ in their responsiveness to cis and trans IL6 signaling. This IL-6 responsiveness is modulated by ATG, and the responsiveness differs between patients with and without previous acute GVHD. Our results therefore suggest that the effect of JAK2/IL-6 targeting in GVHD therapy differs between patient subsets, but further studies are needed to clarify whether the clinical effects of IL-6 targeting therapy will also differ between patient subsets.

Our overall phosphoprofiling results are summarized in Figure 5. Due to the large number of comparisons, the figure only shows differences with a *p*-value ≤ 0.01:CD3^+^CD4^+^ T cells were generally more responsive to IL-6 than CD3^+^CD8^+^ T cells.IL-6-induced STAT3 phosphoresponses were most common, especially STAT3 (Tyr305) phosphorylation in CD3^+^CD4^+^ T cells, whereas mTOR (Ser2448) phosphoresponses differed between patients subsets (i.e., with/without acute GVHD or ATG).There was wide variation between patients with regard to pathway activation, and this variation was at least partly associated with/dependent on both previous ATG prophylaxis and previous acute GVHD.Patients with previous acute GVHD and/or without pretransplant ATG therapy differed from other patients especially with regard to IL-6-induced mTOR phosphoresponses, whereas the upstream AKT phosphoresponses did not differ.Differences between patients seem to involve both classical and IL-6 trans-signaling.It should be emphasized that an alternative experimental approach based on PMA-induced protein kinase C (PKC) activation confirmed that the downstream effects of T cell activation differed between patients with and without previous acute GVHD.

Taken together, our results suggest that the intracellular T cells effects of IL-6 targeting as a strategy for prophylaxis against or treatment of GVHD will differ between patients.

The role of IL-6 in the regulation of inflammation is complex. First, pro-inflammatory effects are often initiated by trans-signaling, whereas anti-inflammatory effects are usually initiated by classical IL-6 signaling [19]. Second, STAT3 exists in four isoforms, the most common and best characterized being the α and β isoforms that seem to have distinct functions [19,28]. The α isoform expresses both the Tyr705 and Ser727 phosphosites, whereas the β isoform lacks the C-terminal Ser727 [28]. Third, STAT3 exists as monomers, homodimers, homotetrameres or heteromeres [19] and can be localized in the nucleus, mitochondria and cytoplasm [29]. Finally, STAT3 phosphorylation is determined by both kinase and phosphatase activity induced by cellular stress/activation [30], and its protein interactions are also regulated by acetylation, methylation and SUMOylation [19]. In the present study, we investigated the effects of trans (i.e., hyper-IL-6, IL-6+sIL-6R) and classical (i.e., IL-6 alone, IL-6+IL-6R+sgp130Fc) IL-6 stimulation on the activating STAT3 (Tyr705) and STAT3 (Ser727) phosphorylation status. We also studied AKT-mTOR signaling because there is crosstalk between AKT-mTOR and the IL-6R-JAK-STAT3 pathway [14,31]. AKT-mTOR is also a pathway downstream to the TCR [32], and it is important in the pathogenesis of GVHD [33].

Most previous clinical and experimental studies of IL-6 in allotransplant recipients have investigated the effects of IL-6 on immune cells early during GVHD; the same is true for studies in animal models, except for one previous animal study of sclerodermous chronic GVHD and for some biomarker studies [1]. We investigated IL-6 responsiveness relatively late (i.e., +90 days post-transplant) because T cell engraftment as well as the degree of immunosuppression vary between patients during the first three months post-transplant. Day +90 represents a time point when T cell engraftment is expected to be relatively stable, the immunosuppressive treatment is reduced and high-risk patients are usually considered for donor lymphocyte infusions after this time point. Furthermore, except for in a few small studies, the effect of IL-6 inhibition has only been investigated during the early post-transplant period; studies of IL-6 responsiveness later after allotransplantation would be of interest when considering IL-6 targeting in the prophylaxis and/or treatment of chronic GVHD [1]. The disadvantage of investigating IL-6 responsiveness at day +90 is that some patients develop late acute GVHD during a further reduction in immunosuppression; analysis of samples collected at this time point thus captures most, but not all, patients developing acute GVHD.

GVHD is a complex multistep process resulting in dysregulation of innate and adaptive immunity, B and T cell development, interactions between host tissue and donor immunocompetent cells, tissue regeneration and crosstalk between the gut microbiome and host gastrointestinal tissue [34]. IL-6 influences a wide variety of biological processes, including innate and adaptive immunity [1]. Hence, IL-6 is expected to influence several steps in the pathogenesis of GVHD and is therefore a therapeutic target in GVHD [1,2,3].

Acute GVHD is associated with increased risk of later chronic GVHD [35,36]. Furthermore, STAT3-phospho-(Tyr705) is increased in circulating CD4^+^ T cells from allotransplant recipients prior to the onset of acute GVHD (20) and seems to induce differentiation of pro-inflammatory Th17 cells and inhibition of Treg development [20,37]. Similarly, animal studies suggest that inhibition of phosphorylated STAT3 reduces Th17 development and thereby alloreactivity [38,39]. However, the mTOR pathway may also facilitate Th17 development, and resistance to mTOR inhibition may be overcome by STAT3 inhibition [20]. In this context, we investigated the immediate/early in vitro signaling of IL-6 on the phosphorylation status of STAT3 and AKT/mTOR in subsets of circulating mononuclear cells (i.e., CD4^+^ and CD8^+^ T cells, non-T mononuclear cells). We investigated constitutive phosphorylation because T cells can show TCR-independent proliferation in response to cytokines alone [40], but IL-6 effects after T cell activation (anti-CD3^+^anti-CD28 exposure) were also examined. These effects were compared for patients with and without previous steroid-requiring acute GVHD, but we only compared the main CD3^+^CD4^+^ and CD3^+^CD8^+^ T cell subsets. Although we did not detect any flow cytometric evidence for further IL-6 signaling heterogeneity within CD4^+^ or CD8^+^ T cell subpopulations, we definitely cannot exclude that IL-6 effects in certain minor CD4^+^/CD8^+^ subsets differ from the general effects on CD4^+^/CD8^+^ main subsets, e.g., for IL-6R expressing naïve versus effector memory T cells [23,24].

In our present study, we investigated both resting and activated T cells, and for T cell activation, we examined PMA and TCR activation (i.e., anti-CD3^+^anti-CD28). TCR activates various downstream mediators/pathways, including protein kinase C (PKC), which is also the target of PMA, and STAT3, which is a common target with IL-6R [32,41]. However, the effects of PMA and TCR activation on PKC differ, especially with regard to their effects on the subcellular localization of PKC [42,43]. Furthermore, we have a focus on four different phosphosites that represent downstream targets for both TCR activation and IL-6 signaling (two STAT3 phosphosites, and AKT-mTOR signaling) [14,31,41]. STAT3 can thus be phosphorylated by both TCR and IL-6R, and the same is true for AKT-mTOR [14,41]. Furthermore, there is bidirectional crosstalk between JAK-STAT and AKT-mTOR signaling [14,31,41,44]. Our activation signals and mediators/phosphosites were selected based on these observations.

Our study showed that IL-6 in combination with CD3/CD28 ligation intensified phosphorylation of STAT3 (both the Ser727 and Y705 phosphosites) and mTOR. In contrast, previous studies have described that IL-6 combined with IL-7 and IL-15 can reduce the TCR signaling threshold for various CD8^+^ T cell subsets (i.e., naïve and memory cells) and thereby influence proliferation [40] as well as CD8^+^ T cell differentiation [45]. The explanation for these differences is perhaps that these previous studies focused on late effects (i.e., proliferation, cytotoxicity, phenotype, survival), whereas we investigated the earliest intracellular effects of IL-6 signaling.

The immunoregulatory effects of IL-6 on CD4^+^ T cells have been described in detail previously. IL-6 enhances Th2 differentiation [46] and inhibits Th1 differentiation as well as IFN-γ production [47], stimulates the differentiation of pro-inflammatory Th17 [48,49] and T follicular helper (Tfh) cells [50], and induces IL-21 production as well as B cell differentiation [51]. In contrast, IL-6 effects on CD8^+^ T cells are less well characterized. However, a recent study [45] showed that IL-6 induces specific CD8^+^ effector T cells with high IL-21 and low IFN-γ release similar to the follicular CD4^+^ T cells, and IL-6 is also important for the development of CD8^+^ cytotoxic T cells [52]. Our present studies suggest that early post-transplant CD4^+^ and CD8^+^ T cells also share similarities with regard to the effects of classic and IL-6 trans-signaling on early intracellular events following TCR activation.

Only naïve T cells and memory T cells express membrane-bound IL-6R, whereas most T cell subsets lack membrane-bound IL-6R and therefore only respond to IL-6 trans-signaling [23,24]. Even though we did not perform a detailed study of IL-6 responsiveness in various CD4^+^/CD8^+^ T cell subsets, it seems justified to conclude that post-transplant circulating CD4^+^ and CD8^+^ T cells derived from patients with and without previous acute GVHD differ in their IL-6 responsiveness. However, more detailed studies of IL-6 effects in various T cell subsets are needed.

Increased STAT3 (Tyr705) phosphorylation after IL-6 stimulation was recently described in patients with autoimmune type 1 diabetes, and this increased responsiveness was seen both for CD4^+^ and CD8^+^ T cells [53]. A similar increase in IL-6 responsiveness in CD4^+^ T cells may also be a contributing factor in the pathogenesis of GVHD [20], but we did not find any evidence of increased IL-6R expression as the mechanism behind the increased IL-6 responsiveness for our patients with previous GVHD. Finally, differences in IL-6 responsiveness were not associated with different levels of circulating Th1/Th17 cells, an observation suggesting that increased IL-6 responsiveness alone is not sufficient for the induction of GVHD-associated Th1/Th17 phenotypes. Differences in T cell subset compartmentalization may also be important, e.g., as an explanation of increased Th1 levels in patients without previous acute GVHD.

ATG is used for GVHD prophylaxis, and its immunosuppressive effects are probably mediated through several mechanisms including cell depletion and altered T cell trafficking [54,55]. Our present study showed that pretransplant ATG treatment was associated both with the absence of certain IL-6 responses (possibly due to altered immunoregulation) and the appearance of new significant responses (possibly due to the depletion of a nonresponsive subset). ATG is a polyclonal antibody preparation that includes antibodies specific for various T cell epitopes as well as antibodies reactive with non-T mononuclear cells [55], and this explains why altered responsiveness is observed both for CD3^+^ T cells and CD3^−^ non-T cells. Such effects were detected both for resting, PMA-activated and TCR-activated cells. We conclude that ATG modulates post-transplant IL-6 responsiveness and may thus influence the post-transplant effects of IL6 targeting.

We could not detect any associations between differences in IL-6 signaling and later chronic GVHD. However, these results should be interpreted with great care because our patient cohort is relatively small and our patients had a relatively short observation time.

Therapeutic targeting of tyrosine kinases downstream to cytokine receptors is a new strategy for immunosuppression and thereby also for GVHD prevention and treatment [56,57,58,59,60,61]. Our present results suggest that early post-transplant CD3^+^CD4^+^ and CD3^+^CD8^+^ T cell subsets differ in their responsiveness to cis and trans IL6 signaling. This phosphoresponsiveness is modulated by ATG and differs between patients with and without previous acute GVHD. Our present studies therefore suggest that IL-6 targeting should be further explored as a strategy for prophylaxis or treatment of late acute and chronic GVHD. However, as explained above, the effects of STAT3 activation are difficult to predict, and additional studies are needed to clarify the molecular mechanisms behind and the functional effects of the altered/increased IL-6 responsiveness in patients with previous acute GVHD. These studies should include more detailed examination of CD4^+^ and CD8^+^ T cell subsets, detailed examination of various STAT3 isoforms, other forms of post-transcriptional STAT3 modifications and studies of intracellular STAT3 compartmentalization. A more detailed characterization of IL-6 effects on various T cell subsets will probably be important for optimal use of IL-6/JAK/STAT3 targeting in allotransplant recipients.

Post-transplant T cell reconstitution varies greatly between individuals and is dependent on GVHD prophylaxis. In individuals treated with in vitro or in vivo T cell depletion (e.g., ATG), adequate numbers of circulation T cells are observed only after several months, while other strategies such as post-transplant cyclophosphamide preserve specific T cell subsets such as regulatory T cells. Furthermore, ATG treatment results in a broader T cell depletion that is dose-dependent and influenced by patient lymphocyte count and/or the amount of immune cells in the infused graft. We observed a significant variation in IL-6 responsiveness with some patients showing a significant response. Although we were not able to identify a specific relationship with GVDH prophylaxis, only a small number of patients treated with post-transplant cyclophosphamide was included.

Following acute GVHD, immunosuppression tapering immunosuppressive therapy is currently guided by clinical signs of GVHD and the estimated risk of infectious complications. Our study showed that IL-6 responsiveness remains higher in patients with previous acute GVHD, with a subset of patients showing significantly high IL-6 responsiveness following GVHD. This might indicate that, for a subset of patients, prolonged immunosuppressive therapy is warranted to sufficiently inhibit alloreactivity.

Our study has several limitations and the results should therefore be interpreted with care. First, we investigated a relatively small group of patients, and the patients were heterogeneous with regard to both pretransplant characteristics and transplantation procedures, but we would emphasize that we investigated unselected patients from a national transplantation center. Our study showed the significant effects of IL-6 stimulation, despite this patient heterogeneity. Second, we investigated a limited number of intracellular key mediators, but there is extensive crosstalk between intracellular signaling pathways and future studies have to clarify whether such crosstalk is important for the IL-6 effects. Third, we observed differences in IL-6 responsiveness between patients with and without previous acute GVHD; it cannot be judged from our present data whether these differences reflect differences predisposing to or induced by the previous acute GVHD. Finally, it is not known whether the present observations are relevant for other transplant recipients, e.g., umbilical cord stem cell transplantation or haploidentical transplantation.

## 5. Conclusions

IL-6 targeting is now used as GVHD prophylaxis and treatment in allotransplant recipients. Our present study shows that IL-6-initiated intracellular signaling can be altered by cis- and trans-signaling both for CD3^+^CD4^+^ and CD3^+^CD8^+^. However, the IL-6 effects varied between patients and were at least partly associated with both previous ATG prophylaxis and previous acute GVHD. Taken together, our observations suggest that future clinical studies should further investigate whether the effects of IL-6 targeting show similar variation between patients and that the use of this immunotherapeutic strategy should be used based on the IL-6 responsiveness of the post-transplant T cell subsets.

## Figures and Tables

**Figure 1 jcm-11-02530-f001:**
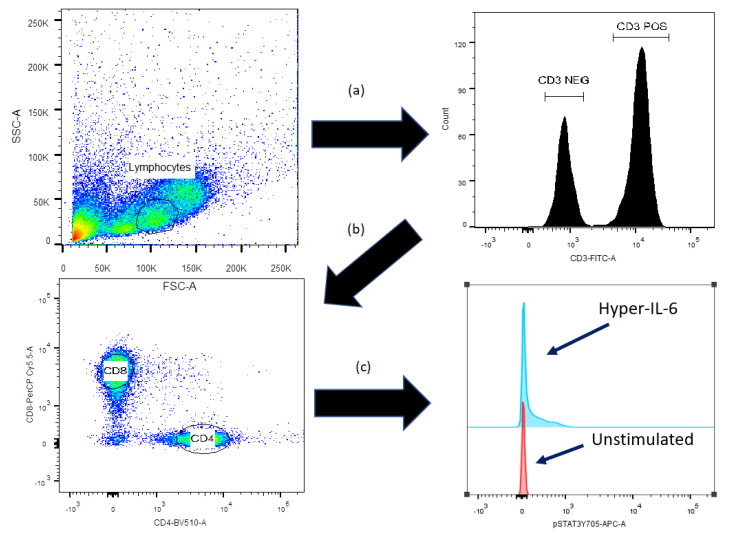
**Overview of the gating strategy for comparison of stimulated and unstimulated cells.** (**a**) After exclusion of duplicates and dead cells from the lymphocyte gate, CD3^+^ and CD3^−^ gates were selected. (**b**) The CD3^+^ cells were thereafter further divided into CD4^+^ and CD8^+^ cells. (**c**) Stimulated samples were finally compared with the unstimulated control cells; this comparison was carried out for the CD3^−^, CD3^+^CD4^+^ and CD3^+^CD8^+^ cells.

**Figure 2 jcm-11-02530-f002:**
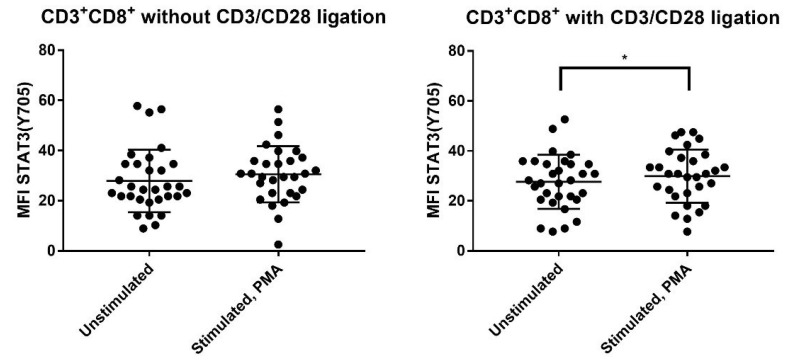
The effects of PMA on STAT3 (Tyr705) phosphorylation of post-transplant CD3^+^CD8^+^ T cells derived from 31 allotransplant recipients on day +90 post-transplant. The T cells were stimulated with PMA either in the presence (**left**) or absence (**right**) of the T cell activation signal anti-CD3^+^anti-CD28. The results are presented as the mean fluorescence intensity (MFI). The Wilcoxon signed-rank test was used for the analyses. * indicates *p*-value 0.01.

**Figure 3 jcm-11-02530-f003:**
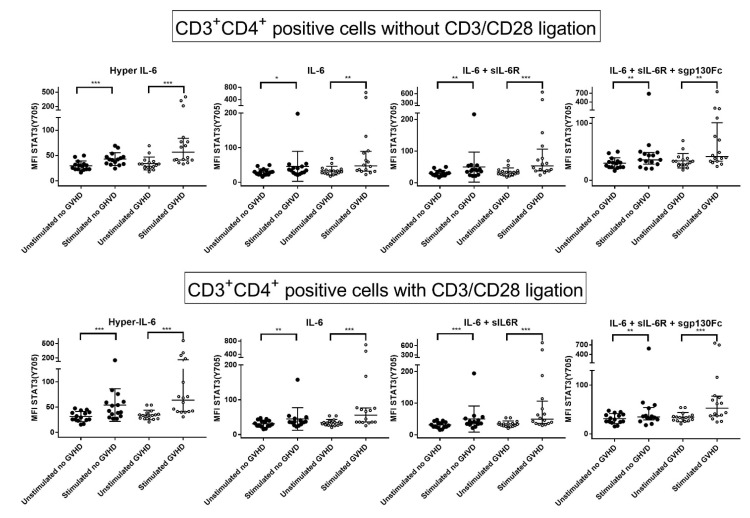
**(page 8).** The effect of classical and trans IL-6 signaling on STAT3 (Tyr705) phosphorylation for post-transplant CD3^+^CD4^+^ T cells derived from 31 allotransplant recipients on day +90 post-transplant; a comparison of patients with and without previous acute GVHD. The T cells were stimulated either with hyper IL-6 (**left**, trans-signaling), IL-6 (**middle left**, classical signaling), IL-6+IL-6R (**middle right**, classical and trans-signaling) and IL-6+sIL-6R+sgp130FC, classical signaling). IL-6 stimulation was tested in the absence (**upper part**) and in the presence (**lower part**) of the T cell activation signal anti-CD3^+^anti-CD28. The results are presented as the mean fluorescence intensity (MFI), and for each of the four IL-6 stimulations, we compared patients without (dark symbols) and with previous acute GVHD (open symbols). The Wilcoxon signed-rank test was used for the statistical analyses. * indicates a *p*-value 0.05 to 0.01, ** indicates a *p*-value 0.001 to 0.001, *** indicates *p*-value < 0.001.

**Figure 4 jcm-11-02530-f004:**
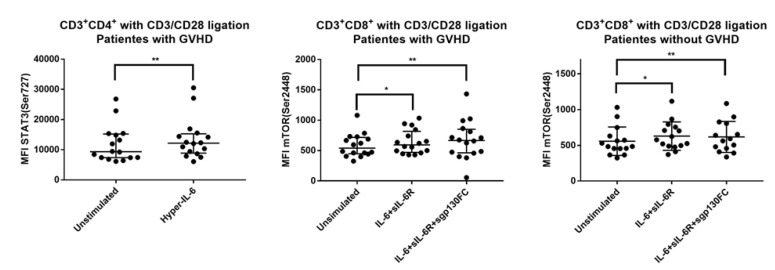
The effect of IL-6 stimulation on intracellular signaling in TCR-activated T cells derived from 31 allotransplant recipients on day +90 post-transplant. The T cells were stimulated either with hyper IL-6 (trans-signaling), IL-6+sIL-6R (classical and trans-signaling) or IL-6+sIL-6R+sgp130FC (classical signaling) in the presence of the T cell activation signal anti-CD3^+^anti-CD28. The results are presented as the mean fluorescence intensity (MFI), and unstimulated refers to no IL-6 stimulation but only anti-CD3^+^anti-CD28 activation. The figure presents the results for (**left**) effects on STAT3 (Ser727) phosphorylation of CD3^+^CD4^+^ T cells derived from patients with GVHD stimulated with hyper-IL-6; (**center**) effects on mTOR (Ser2448) phosphorylation in CD3^+^CD8^+^ T cells derived from patients with GVHD and stimulated with IL-6+sIL-6R or IL-6+sIL-6R+sgp130FC; and (**right**) effects on mTOR (Ser2448) phosphorylation in CD3^+^CD8^+^ T cells derived from patients without GVHD and stimulated with IL-6+sIL-6R or IL-6+sIL-6R+sgp130FC. The Wilcoxon signed-rank test was used for the analyses. * indicates a *p*-value 0.05 to 0.01, ** indicates a *p*-value < 0.01.

**Figure 5 jcm-11-02530-f005:**
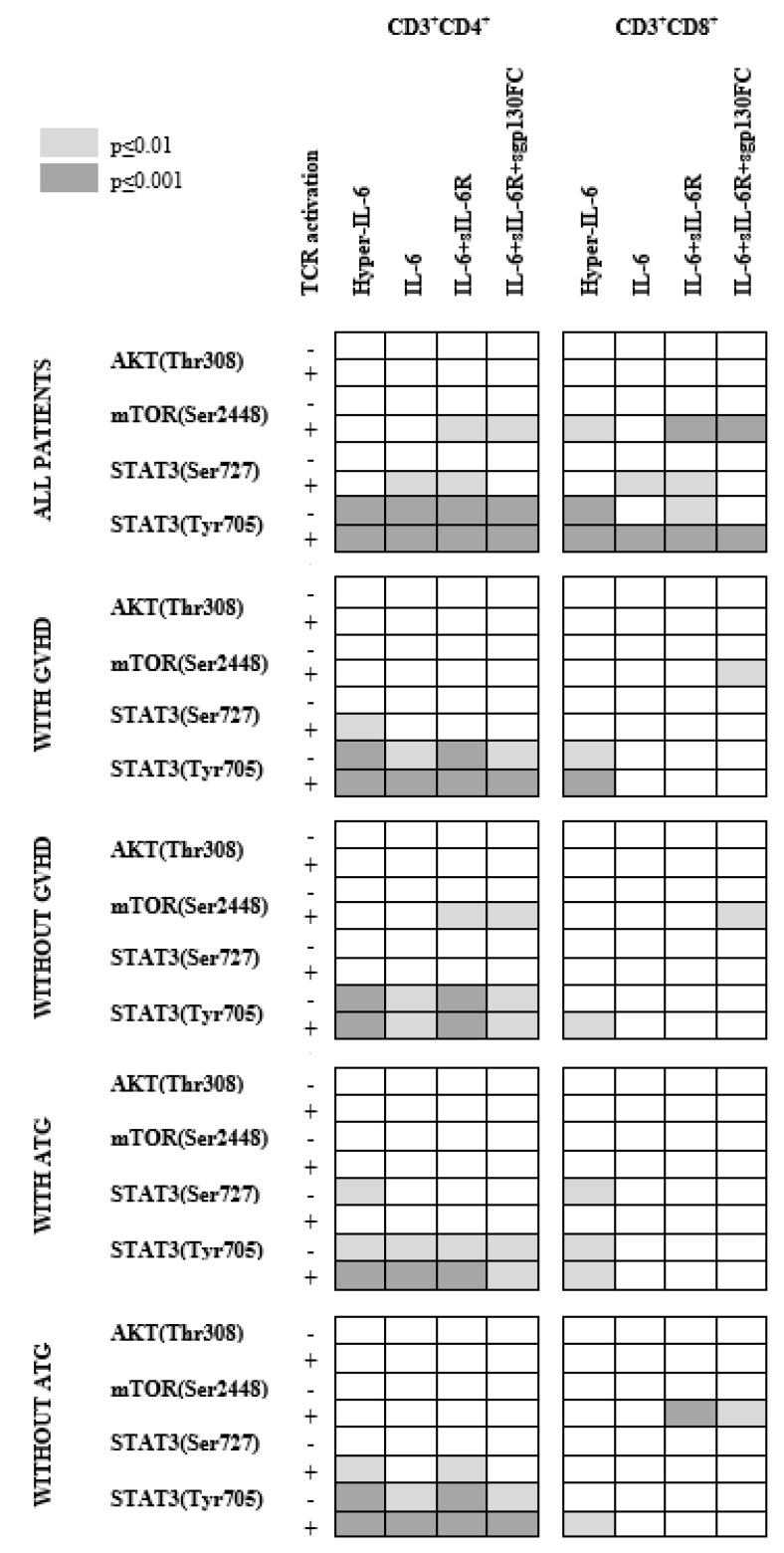
The IL-6 responsiveness of CD3^+^CD4^+^ and CD3^+^CD8^+^ T cells derived from allotransplant recipients at day +90 post-transplant. We investigated the effects of four different forms of IL-6 stimulation on the T cell subsets: IL-6 alone (classical IL-6 signaling), hyper-IL-6 (trans-signaling), IL-6 plus sIL-6R (classical and trans-signaling) and IL-6+sIL-6R+sgp130FC (classical Il-6 signaling; trans-signaling blocked by sgp130FC). We investigated the effects of these signals on AKT (Thr308), mTOR (Ser2448), STAT3 (Ser727) and STAT3 (Tyr705) phosphorylation in resting cells and in TCR-activated cells (anti-CD3^+^anti-CD28). We analyzed the overall results for the 31 patients. All statistical analyses were carried out using the Wilcoxon signed-rank test. The figure summarizes our overall results (see Table 3 and Table 4), but due to the overall number of comparisons, we only present differences with a *p*-value < 0.01.

**Table 1 jcm-11-02530-t001:** The characteristics of the allotransplant recipients and their donors included in the analysis.

Recipient Characteristics (*n* = 31)	
Age, median and range (Years)	59 (21–72)
Gender, Female/Male	11/20
**Diagnosis (number)**	
AML, de novo	11
Myelodysplastic syndrome, high-risk	6
Acute lymphoblastic leukemia	3
Chronic myeloid leukemia, myelofibrosis, chronic myelomonocytic leukemia, myeloproliferative neoplasia unspecified	1 of each
Aplastic anemia	2
**Conditioning regimes (number)**	
Busulfan + cyclophosphamide (myeloablative condition)	4
Fludarabine + busulfan (reduced intensity conditioning)	13
Fludarabin + Treosulfan	7
Fludarabin + cyclophosphamide	5
Total body irradiation (TBI) + cyclophosphamide	1
Fludarabin + Thiotepa BU	1
**GVHD prophylaxis (number)**	
Cyclosporine A + methotrexate	16
Cyclosporine A + methotrexate + antithymocyte globulin	10
Cyclosporine A + sirolimus	1
Cyclosporine A + sirolimus + ATG	3
Post-transplant cyclophosphamide	1
**Stem cell source (number)**	
Peripheral blood mobilized stem cells	23
Bone marrow grafts	8
**Ongoing immunosuppression on day +90 post-transplant (number)**	
Cyclosporine	31
Corticosteroids	5
Others	3
**Ongoing immunosuppression and alive on day +360 post-transplant (number)**	
Cyclosporine	6
Corticosteroids	4
Others	4
**Donor Characteristics**	
Fully matched sibling/Matched unrelated donor/Haploidentical relative	13/17/1

The indications for high-dose steroid treatment were acute GVHD grade II with gastrointestinal involvement or Grade III/IV acute GVHD.

**Table 2 jcm-11-02530-t002:** The effects of PMA stimulation on circulating CD3^+^CD4^+^ T cells, CD3^+^CD8^+^ T cells and CD3^−^ mononuclear cells derived from allotransplant recipients on day +90 post-transplant. Cells were stimulated either with PMA alone or with PMA combined with anti-CD3^+^anti-CD28, and the effect of these stimulations on the phosphorylation of AKT/mTOR/STAT3 was evaluated by flow cytometry. All *p*-values refer to a statistical comparison (i.e., increased phosphorylation) between cultures with an activation signal (PMA alone or PMA+anti-CD3^+^anti-CD28) and the corresponding control cultures prepared in medium alone. All comparisons were carried out using the Wilcoxon signed-rank test and we only present stimulations with a *p*-value < 0.05.

	CD3^+^CD4^+^ T Cells		CD3^+^CD8^+^ T Cells		CD3^−^
Phosphotarget	PMA Alone	PMA Anti-CD3 Anti-CD28	PMA Alone	PMA Anti-CD3 Anti-CD28	PMA Alone
**All patients**					
AKT (Thr308)	<0.0001	<0.0001	<0.0001	<0.0001	<0.0001
mTOR (Ser2448)	<0.0001	<0.0001	<0.0001	<0.0001	<0.0001
STAT3 (Ser727)	<0.0001	<0.0001	<0.0001	<0.0001	<0.0001
STAT3 (Tyr705)				0.03	
**Patients with previous acute GVHD (*n* = 16)**					
AKT (Thr308)	<0.0001	0.0007	0.0009	<0.0001	0.02
mTOR (Ser2448)	0.002	0.0003	<0.0001	0.0006	0.001
STAT3 (Ser727)	<0.0001	0.008	0.0006	0.04	0.0003
STAT3 (Tyr705)					
**Patients without previous acute GVHD (*n* = 15)**					
AKT (Thr308)		0.02	.008	0.002	
mTOR (Ser2448)	<0.0001	<0.0001	<0.0001		<0.0001
STAT3 (Ser727)		0.02	0.03	0.004	
STAT3 (Tyr705)					

**Table 3 jcm-11-02530-t003:** The effects of four different forms of IL-6 stimulation on CD3^+^CD4^+^ circulating T cells. The effects on the phosphorylation of the intracellular mediators STAT3/Akt/mTOR were investigated. IL-6 stimulation was tested without and with the activation signal anti-CD3^+^anti-CD28. The table presents the statistically significant increases, i.e., *p*-values < 0.05. The T cells were derived from the peripheral blood of allotransplant recipients with and without previous acute GVHD on day +90 post-transplant. All comparisons were carried out using the Wilcoxon signed-rank test.

	Without TCR Ligation				With TCR Ligation			
Phosphotarget	Hyper- IL-6	IL-6	IL-6 sIL-6R	IL-6 sIL-6R sgp130	Hyper-IL-6	IL-6	IL6 sIL-6R	IL-6 sIL-6R sgp130
**All patients**								
AKT (Thr308)								
mTOR (Ser2448)							0.009	0.0056
STAT3 (Ser727)				0.03	0.02	0.007	0.007	
STAT3 (Tyr705)	<0.0001	<0.0001	<0.0001	<0.0001	<0.0001	<0.0001	<0.0001	<0.0001
**Patients with GVHD (*n* = 16)**								
AKT (Thr308)								
mTOR (Ser2448)								0.04
STAT3 (Ser727)					0.009			
STAT3 (Tyr705)	<0.0001	0.003	0.0008	0.004	<0.0001	0.0002	0.0003	0.0008
**Patients without GVHD (*n* = 15)**								
AKT (Thr308)								
mTOR (Ser2448)							0.009	0.006
STAT3 (Ser727)								
STAT3 (Tyr705)	<0.0001	0.002	0.001	0.004	<0.0001	0.002	0.0005	0.006

**Table 4 jcm-11-02530-t004:** The effects of four different forms of IL-6 stimulation on circulating CD3^+^CD8^+^ T cells. We investigated effects on the phosphorylation of the intracellular mediators STAT3/Akt/mTOR. IL-6 stimulation was tested alone and in the presence of the activation signal anti-CD3^+^anti-CD28. The table presents the statistically significant increases, i.e., *p*-values < 0.05. The T cells were derived from the peripheral blood of allotransplant recipients with and without previous acute GVHD on day +90 post-transplant. All comparisons were carried out using the Wilcoxon signed-rank test.

	Without TCR Ligation				With TCR Ligation			
**Phosphotarget**	Hyper- IL-6	IL-6	IL-6 sIL-6R	IL-6 sIL-6R sgp130	Hyper- IL-6	IL-6	IL6 sIL-6R	IL-6 sIL-6R sgp130
**All patients**								
AKT (Thr308)							0.03	
mTOR (Ser2448)			0.03		0.005	0.04	<0.001	<0.001
STAT3 (Ser727)					0.03	0.003	0.004	
STAT3 (Tyr705)	<0.0001	0.03	0.004		<0.0001	<0.0001	<0.0001	<0.0001
**Patients with GVHD (*n* = 16)**								
AKT (Thr308)								
mTOR (Ser2448)							0.04	0.008
STAT3 (Ser727)			0.04					
STAT3 (Tyr705)	0.003				0.0008			
**Patients without GVHD (*n* = 15)**								
AKT (Thr308)								
mTOR (Ser2448)							0.03	0.005
STAT3 (Ser727)							0.02	
STAT3 (Tyr705)					0.002			

**Table 5 jcm-11-02530-t005:** Levels of circulating T cell subsets in allotransplant recipients. We investigated 24 patients and present the results for all patients and patients with/without previous acute GVHD. Different groups were compared using the Mann–Whitney-U test. IL-6R expression is presented as the mean fluorescence intensity.

	All Patients Median (Range)	Patients with Previous Acute GVHD	Patients without Previous Acute GVHD	*p*-Value GVHD vs. No GVHD
**Circulating T cell subset**				
CD3^+^CD4^+^(% of CD3^+^ cells)	42 (10–73)	50 (10–73)	36 (18–54)	0.06
CD3^+^CD8^+^(% of CD3^+^ cells)	48 (22–83)	44 (22–83)	51 (35–79)	0.18
Th1 (% of CD4^+^ T cells)	11 (2–38)	9 (5–19)	15 (2–38)	0.003
Th2 (% of CD4^+^ T cells)	5 (0.9–11)	4.6 (1.5–9)	4 (0.9–11)	0.86
Th17 (% of CD4^+^ T cells)	0.9 (0.3–3.0)	0.8 (0.3–3.0)	1 (0.3–1.7)	0.43
**IL-6R expression**				
CD3^+^CD4^+^	241 (160–420)	267 (160–420)	223 (191–383)	0.43
CD3^+^CD8^+^	170 (134–261)	179 (134–261)	169 (142–226)	0.37
CD3^−^	225 (194–366)	227 (194–279)	225 (204–336)	0.97

## Data Availability

The raw data supporting the conclusions of this article will be made available by the authors, without undue reservation.

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
