# Peer review of "IL-6 Responsiveness of CD4+ and CD8+ T Cells after Allogeneic Stem Cell Transplantation Differs between Patients and Is Associated with Previous Acute Graft versus Host Disease and Pretransplant Antithymocyte Globulin Therapy"

_jcm, 2022, doi:10.3390/jcm11092530_

Round 1

Reviewer 1 Report

This article from Tor Henrik Anderson Tvedt et al. describes the effects of IL-6 signaling on the mTOR and JAK/STAT pathways in T cells after allogeneic stem cell transplantation, and provides an interesting contribution to unravel the complexity of this signaling. Using primary cells instead of cell line models adds to this work. Some comments:

- In Table 1: the total number of patients in the 'Diagnosis', 'Conditioning regimes' and 'Stem cell source' section doesn't reach 31. If there's some data missing, it should be displayed.

- Tables 2, 3 and 4: the number of samples from patients without/with acute GVHD should be explicit, in the tables or in their legend. The number of asterisk associated with p-value should be homogenized between Figures 2, 3 and 4, to avoid any confusion. Besides, in some cases, it seems only a few samples has a significant increase of expression after stimulation on the dot plots; do the authors have any hypothesis(es) about the clinical or biological characteristics of these samples?

- Figure 2: the two graphs could be swapped, to keep the order of the columns in the table above.

- The Supplementary Table 4 doesn't seem to be referred to in the text.

- Some minor spelling corrections:

. line 87: naive T cells lose IL-6R expression

. line 97: ligation on circulating T cells

. Table 3: .006 (instead of ,006)

. line 313: STAT3(Tyr705) or (Y705)

. line 314: the two patient subsets

. line 317: no dot before "Figure 4"

. line 450: what should be emphisized?

Author Response

We are grateful for the reviewer comments. We have done our best to address all these comments, and our response is given below. All changes made to the manuscript have been marked in yellow.

Response to reviewer 1:

  1. In Table 1: the total number of patients in the 'Diagnosis', 'Conditioning regimes' and 'Stem cell source' section doesn't reach 31. If there's some data missing, it should be displayed.

Comments from the authors:

We appreciate this comment, this was a typing error and the table has now been updated.

  1. Tables 2, 3 and 4: the number of samples from patients without/with acute GVHD should be explicit, in the tables or in their legend. The number of asterisk associated with p-value should be homogenized between Figures 2, 3 and 4, to avoid any confusion. Besides, in some cases, it seems only a few samples has a significant increase of expression after stimulation on the dot plots; do the authors have any hypothesis(es) about the clinical or biological characteristics of these samples?

Comments from the authors:

We appreciate this comment and this has now been changed. The exact number of patients with and without GVHD is now explicitly described in the tables and the figure legends.

We have now added a short discussion on possible effects on various types of GVHD prophylaxis.

  1. Figure 2: the two graphs could be swapped, to keep the order of the columns in the table above.

Comments from the authors:

The two graphs have now been swapped.

  1. The Supplementary Table 4 doesn't seem to be referred to in the text.

Comments from the authors:

This table is referred to in line 430, although no changes have been made, the section is highlighted in yellow.

  1. Some minor spelling corrections:

Comments:

These spelling errors have now been corrected. However, we were not able to identify the errors: line 317: no dot before "Figure 4" and. line 450: what should be emphasized?”, as the line number does not seem to correspond with the text citations.

Reviewer 2 Report

Interesting data on different cellular response to IL-6 signaling

Despite low patient number, did you observe any difference in cellular responses between patients with responsive GvHD and patients with refractory GvH?

Patients were heterogeneous in term of GvHD prophylaxis. did you observe / do you expect any difference in IL-6 responsiveness based on previous GvHD ppx (other than ATG)?

I believe the authors should provide more comments on how these data would, potentially, guide clinical practice.

Author Response

Response to reviewer 2:

  • Did you observe any difference in cellular responses between patients with responsive GvHD and patients with refractory GvHD

Comments:

Unfortunately no patients with pre day-90 severe steroid refractory acute GVHD were included in the study. As described in section 3.13 and line 617-640, we were not able to identify any correlation between T cells responses and later development of chronic GVHD or possibility to wean off immunosuppressive therapy.

  • Patients were heterogeneous in term of GvHD prophylaxis. did you observe / do you expect any difference in IL-6 responsiveness based on previous GvHD ppx (other than ATG).

Comments:

We have now added a short section were we discuss potential differences in IL-6 responsiveness with various types of GVHD prophylaxis

  • I believe the authors should provide more comments on how these data would, potentially, guide clinical practice.

Comments:

We have now added a short section on possible implications of our findings.